# Muscle Lipid Oxidation Is Not Affected by Obstructive Sleep Apnea in Diabetes and Healthy Subjects

**DOI:** 10.3390/ijms24065308

**Published:** 2023-03-10

**Authors:** Zuzana Lattova, Lucie Slovakova, Andrea Plihalova, Jan Gojda, Moustafa Elkalaf, Katerina Westlake, Jan Polak

**Affiliations:** 1Department of Pathophysiology, Third Faculty of Medicine, Charles University, 100 00 Prague, Czech Republic; zuzanalattova@gmail.com (Z.L.); lucie.slovakova@lf3.cuni.cz (L.S.); moustafa.elkalaf@gmail.com (M.E.); katerina.westlake@lf3.cuni.cz (K.W.); 2Department of Internal Medicine, University Hospital Kralovske Vinohrady, 100 34 Prague, Czech Republic; plihalova.andrea@gmail.com (A.P.); jan.gojda@lf3.cuni.cz (J.G.); 3Centre for Research on Diabetes, Metabolism and Nutrition, Third Faculty of Medicine, Charles University, 100 00 Prague, Czech Republic

**Keywords:** obstructive sleep apnea, type 2 diabetes mellitus, free fatty acids, lipid utilization, IVGTT, glucose intolerance, muscle metabolism, hypoxia

## Abstract

The molecular mechanisms linking obstructive sleep apnea (OSA) with type 2 diabetes mellitus (T2DM) remain unclear. This study investigated the effect of OSA on skeletal muscle lipid oxidation in nondiabetic controls and in type 2 diabetes (T2DM) patients. Forty-four participants matched for age and adiposity were enrolled: nondiabetic controls (control, *n* = 14), nondiabetic patients with severe OSA (OSA, *n* = 9), T2DM patients with no OSA (T2DM, *n* = 10), and T2DM patients with severe OSA (T2DM + OSA, *n* = 11). A skeletal muscle biopsy was performed; gene and protein expressions were determined and lipid oxidation was analyzed. An intravenous glucose tolerance test was performed to investigate glucose homeostasis. No differences in lipid oxidation (178.2 ± 57.1, 161.7 ± 22.4, 169.3 ± 50.9, and 140.0 ± 24.1 pmol/min/mg for control, OSA, T2DM, and T2DM+OSA, respectively; *p* > 0.05) or gene and protein expressions were observed between the groups. The disposition index, acute insulin response to glucose, insulin resistance, plasma insulin, glucose, and HBA_1_C progressively worsened in the following order: control, OSA, T2DM, and T2DM + OSA (*p* for trend <0.05). No association was observed between the muscle lipid oxidation and the glucose metabolism variables. We conclude that severe OSA is not associated with reduced muscle lipid oxidation and that metabolic derangements in OSA are not mediated through impaired muscle lipid oxidation.

## 1. Introduction

Obstructive sleep apnea (OSA) syndrome is an often-neglected disorder with a prevalence of 5–15% in the general population of adults that increases to 50–80% in type 2 diabetes (T2DM) and obese patients [1,2], increasing the cardiovascular and all-cause mortality of affected individuals [3,4]. Based on the clinical features of transient upper airway collapse during sleep, causing repetitive oxyhemoglobin desaturations (intermittent tissue hypoxemia) and sleep fragmentation [5], cross-sectional and prospective studies have identified OSA as a risk factor for the development of glucose intolerance, insulin resistance, and T2DM that was independent of other established risk factors (e.g., age, genetic factors, obesity, and physical inactivity) [6,7,8].

The causal link between OSA and T2DM remains under active investigation, employing in vitro, animal, and human experiments. Despite the methodological challenges hampering in vitro hypoxic studies [9], independent groups have reported using intermittent hypoxia (mimicking the oxygen desaturations present in OSA patients) to reduce insulin signaling in hepatocytes [10] and adipocytes [11], which leads to reduced glucose uptake and augmented lipolysis [12]. Exposing rodents to intermittent hypoxia caused insulin resistance, glucose intolerance, and β-cell dysfunction, and led to increased hepatic glucose output, hyperglycemia, and increased adipose tissue lipolysis in mice [13,14,15,16,17,18,19]. Identical exposure also induced hyperglycemia and a reduced expression of muscle glucose transporters in rats [20,21]. Furthermore, experiments in healthy humans have demonstrated insulin resistance, impaired β-cell function, and elevated glucose levels after acute exposure to intermittent hypoxia [22,23].

Suggested molecular and endocrine mechanisms that could alter the glucose homeostasis in OSA include increased sympathetic activity, inflammatory pathway stimulation, reactive oxygen species generation, elevated plasma corticoid or endothelin-1 levels, or a modified adipokine secretion profile [8,24]. Additionally, an important role for hypoxia-inducible factors (HIF) has also been identified and summarized [25]. More recently, increased levels of circulating free fatty acids (FFAs) have attracted attention as a potential link between hypoxia and impaired glucose metabolism [26,27]. This is due to their ability to impair the glucose uptake and metabolism in muscle through the inhibition of critical glycolytic enzymes [28,29]; i.e., to induce insulin resistance in the liver through modified intracellular signaling and altered gene expression (resulting in excessive hepatic glucose output) [30,31,32], as well as to decrease insulin secretion and stimulate apoptosis in β cells [33,34]. The balance between FFA release into the circulation (adipose tissue lipolysis) and its uptake in the peripheral tissues (oxidation and/or storage) determines the actual plasma FFA concentration [35,36,37]. Previous in vitro [12], rodent [19], and human [26,38] studies have shown that low pericellular oxygen levels upregulate lipolysis, which is also upregulated in OSA patients. In contrast, the crucial factor of net plasma FFA balance and its uptake and utilization in the peripheral tissues under intermittent hypoxia remains to be elucidated.

This study investigated whether OSA modifies skeletal muscle oxidation in otherwise healthy participants and T2DM patients. To achieve this goal, we thoroughly characterized a host of metabolic features (using an intravenous glucose tolerance test) and investigated lipid oxidation in muscle biopsies obtained from nondiabetic as well as T2DM patients diagnosed with severe OSA. We then compared them with matched nondiabetic and T2DM controls (with absent or mild OSA). The outcomes of this study could provide a basis for pharmacological or nonpharmacological (exercise) interventions targeting muscle lipid oxidation for treating the metabolic complications associated with OSA.

## 2. Results

### 2.1. Anthropometric, Biochemical, and Sleep Characteristics

A summary of the basic demographic, anthropometric, metabolic, and sleep characteristics of all the participants is presented in Table 1. The data show successful age, BMI, and adiposity matching between the groups. As expected from the recruitment protocol, the groups affected by severe OSA showed higher indices of sleep-disordered breathing e.g. apnea-hypopnea index (AHI) or oxygen desaturation index (ODI) and spent more time in hypoxia during the night than did the participants without OSA. The presence of OSA worsened whole-body glucose homeostasis, expressed as intravenous glucose tolerance test-derived disposition index (DI), by 39% and 51% in nondiabetic and T2DM diabetes patients, respectively. Progressive worsening of the metabolic variables including DI, acute response to insulin (AIRg), insulin resistance, HOMA-IR (homeostatic model assessment for insulin resistance), fasting insulin, fasting glucose, and HBA_1_C (glycated hemoglobin) was observed (*p* for trend <0.05, mixed-model ANOVA). No differences in the plasma FFA levels were observed between the groups.

### 2.2. Oxygen Consumption Rate and Substrate Utilization in Skeletal Muscle Biopsy

To investigate the ability of the skeletal muscle cells to utilize palmitate as an energy source, quadricep muscle biopsies were assessed ex vivo with added energy substrates using direct respirometry. As shown in Figure 1A, the spontaneous (basal) O_2_ consumption rate (OCR) did not differ between the groups. Subsequently, palmitoyl carnitine was added to the incubation buffer to quantify palmitate-induced respiration. Although the addition of the palmitate stimulated respiration by 52%, 38%, 30%, and 56% in control (117.3 ± 33.3 vs. 178.2 ± 57.1 pmol/min/mg, *p* < 0.05), OSA (117.5 ± 16.1 vs. 161.7 ± 22.4 pmol/min/mg, *p* < 0.05), T2DM (130.5 ± 45.5 vs. 169.3 ± 50.9 pmol/min/mg, *p* < 0.05), and T2DM + OSA (89.7 ± 14.3 vs. 140.0 ± 24.1 pmol/min/mg, *p* < 0.05), respectively, no differences in OCR in response to palmitate administration were observed between the groups (ANOVA, *p* > 0.05) (Figure 1B). Similar responses were demonstrated after the administration of succinate, which increased oxygen consumption by 116%, 172%, 137%, and 80% in control (178.2 ± 57.1 vs. 385.5 ± 140 pmol/min/mg, *p* < 0.05), OSA (161.7 ± 22.4 vs. 441.1 ± 51.0 pmol/min/mg, *p* < 0.05), T2DM (169.3 ± 50.9 vs. 400.5 ± 140.1 pmol/min/mg, *p* < 0.05), and T2DM + OSA (140.0 ± 24.1 vs. 251.4 ± 39.1 pmol/min/mg, *p* < 0.05), respectively; however, the magnitude of this response was the same between the groups (ANOVA, *p* > 0.05) (Figure 1C). While palmitate-induced respiration was not associated with any of the analyzed anthropometric or biochemical variables, succinate-induced respiration was positively associated with whole-body CO_2_ production (r = 0.361, *p* < 0.05) and, to a lesser extent, with oxygen consumption (r = 0.309, *p* = 0.063) and resting metabolic rate (r = 0.315, *p* = 0.058) as determined with indirect calorimetry.

### 2.3. Protein Expression

The protein expression of the fatty acid plasma membrane and intracellular transporters (fatty acid transport protein 4, FATP4 and platelet glycoprotein 4, CD36) as well as the critical regulator of fatty acid oxidation in the mitochondria (carnitine-palmitoyl transferase I, CPT1) were determined. Protein expression of FATP4 was not affected by OSA in nondiabetic persons (ratio of FATP4 to β-tubulin band intensity: 3.8 ± 1.1 vs. 4.1 ± 1.9, NS) or in diabetic patients (ratio of FATP4 to β-tubulin band intensity: 2.1 ± 0.9 vs. 4.8 ± 2.0, NS). Similarly, the protein expression of CD36 and CPT1 was stable across the control, OSA, T2DM, and T2DM + OSA groups (ratio of CD36 to β-tubulin band intensity: 1.4 ± 0.4, 2.1 ± 0.7, 1.4 ± 0.3, and 2.4 ± 0.8, respectively, all were NS; the ratio of CPT1 to β-tubulin band intensity: 0.3 ± 0.1, 0.4 ± 0.1, 0.3 ± 0.2, and 0.7 ± 0.3, respectively, all were NS) (Figure 2A–C).

The severity of hypoxia, expressed as time spent sleeping with hemoglobin saturation under 90% or 85%, was positively associated with protein expression of CD36 (r = 0.328 and r = 0.321, respectively, *p* < 0.05). FATP4 expression was negatively associated with insulin resistance, expressed as the HOMA-IR index (r = −0.335, *p* < 0.05). Additionally, the protein expression of CPT1 correlated with the protein expression of FATP4 (r = 0.519, *p* < 0.001).

### 2.4. Gene Expression

The gene expression analysis of the skeletal muscle biopsies focused on the genes coding for the FATP4, CD36, and CPT1 proteins. No differences between the groups were detected in the relative gene expression of (1) *FATP4* (0.86 ± 0.06, 0.8 ± 0.07, 0.82 ± 0.05, and 0.79 ± 0.03 for control, OSA, T2DM, and T2DM + OSA groups, respectively, all *p* > 0.05), (2) *CD36* (41.6 ± 3.1, 40.1 ± 4.0, 39.1 ± 2.7, and 36.3 ± 3.4, for control, OSA, T2DM, and T2DM + OSA groups, respectively, all *p* > 0.05), and (3) *CPT1* (0.076 ± 0.007, 0.070 ± 0.008, 0.085 ± 0.010, and 0.080 ± 0.006, for the control, OSA, T2DM, and T2DM + OSA groups, respectively, all *p* > 0.05). *CPT1* expression was negatively associated with Sg (r = −0.169, *p* < 0.05) and GEZI (r = −0,237, *p* < 0.05), while no association was observed between the gene expressions of *FATP4, CD36*, or *CPT1* and the anthropometric, sleep-related, and biochemical variables. The results are shown in Figure 3A–C.

## 3. Discussion

The present study investigated whether OSA syndrome affects skeletal muscle lipid oxidation in matched healthy controls and T2DM patients and is associated with whole-body glucose metabolism parameters. An ex vivo analysis of palmitate oxidation as well as the protein and gene expressions of the key players in fatty acid transport/metabolism (FATP4, CD36, and CPT1) in muscle biopsies demonstrated unequivocally no effect of OSA on lipid utilization or the expression of crucial related proteins. Similarly, no association between muscle lipid oxidation and whole-body insulin sensitivity, insulin secretion, or DI was observed, although the FATP4 protein was negatively associated with insulin resistance (HOMA-IR index).

The molecular mechanisms linking OSA with impaired glucose metabolism and increased risk of T2DM remain only partially elucidated [8]. However, elevated plasma FFA levels have attracted attention owing to their demonstrated role in the pathogenesis of insulin resistance and β-cell dysfunction [39,40,41]. Additionally, studies have demonstrated elevated plasma FFA in patients with OSA [38,42,43], particularly during the night (sleep) hours. Considering adipose tissue lipolysis (the source of circulating FFA) and skeletal muscle FFA utilization as the key factors determining fasting FFA levels (with minor contributions from liver FFA metabolism and FFA oxidation by other organs), this study focused on muscle lipid oxidation as a potential mechanism contributing to elevated plasma FFA levels and, thus, metabolic dysregulation. Impaired lipid oxidation in muscle cells could lead to elevated FFA, resulting in an accumulation of intracellular lipid-derived molecules (e.g., ceramides, phospholipids, and diacylglycerol) that have potent signaling properties and could eventually lead to muscle insulin resistance. The trio of investigated molecules, CD36, FATP4, and CPT1, represent key players in fatty acid transport from the extracellular to the intracellular space (regulated mainly by CD36 and FATP4) and subsequently in FFA transport to the mitochondria for oxidation (secured by CPT1). The dominant role of CD36 in FFA transport can be demonstrated by the reduced transmembrane FFA transport after chemical CD36 inhibition [44,45]; unfortunately, much less is currently known about the role and significance of FATP4 in the transport of FFA across the sarcolemma [46]. Furthermore, CD36 has been suggested to work in concert with CPT1 in mediating mitochondrial FFA transport/oxidation. Importantly, obesity, insulin resistance, and type 2 diabetes [47] have been associated with increased FFA muscle plasma membrane transport and elevated CD36 levels [47,48]. They are also combined with reduced mitochondrial FFA oxidation [28,49,50], ultimately leading to increased accumulation of lipids in the muscle tissue. Hence, an overexpression of CPT1 in muscle improved the high-fat-diet-induced insulin resistance in rats [51,52]. For thorough reviews of the key players in muscle FFA transport and their consequences, readers are referred to [52,53,54].

The unique feature of the present study is that all the groups were matched for adiposity-related variables and age, which enabled the assessment of the contribution of OSA to glucose metabolism impairment without obvious confounding factors. Despite striking differences in glucose metabolism indices, time spent in hypoxia, and the number of apneic/hypopneic episodes, no differences in skeletal muscle palmitate oxidation capacity were observed between the patients with and without severe OSA, which was further corroborated by the unchanged protein and gene expressions of the essential proteins involved in FFA cellular transport and oxidation, i.e., CD36, FAPT4, and CPT1.

Our data suggest that lipid oxidation in the skeletal muscle is not affected by OSA in humans and is not a significant determinant of impaired whole-body glucose metabolism associated with OSA. This conclusion does not conflict with previous in vitro studies reporting decreased FFA uptake and reduced FATP4 and CD36 protein expression after exposure to prolonged severe hypoxia [55] (1% O_2_ for multiple days). Muscle tissue O_2_ levels during severe OSA were shown to reach ~25 mm Hg (~5%) in a mouse model of OSA [56], which is considerably milder hypoxia than that used in in vitro studies. Similarly, diminished FFA oxidation in muscle was observed as a consequence of environmental hypoxia (high-altitude exposure) [57]; however, it should be noted that the hemoglobin desaturation at high altitude is greater (e.g., 54% at an altitude of 8400 m [58]) and lasts significantly longer (days vs. minutes) than that used in the clinical setting of OSA [59,60]. Our results complement a recent study showing (in agreement with the present study) that OSA worsens glucose homeostasis. However, other authors also observed increased intra- and extramyocellular lipid content in skeletal muscle [61]. An increased triglyceride accumulation and modified lipidomic profile in myocytes after hypoxic exposure has been documented in vitro, presumably partly due to de novo lipogenesis [59,60]. These studies suggest a hypothetical picture of unchanged intracellular lipid oxidation combined with increased lipid synthesis as a consequence of hypoxic exposure. Further studies are needed to address and quantify in detail the plasma membrane FFA transport under hypoxic conditions because extrapolations from gene/protein data on the expression of essential protein transporters are rather imprecise.

Rapid changes in plasma FFA levels in response to hypoxia/reoxygenation have been described in chronic obstructive pulmonary disease patients [27] and after oxygen administration in OSA patients [43]. Adipose tissue lipolysis, as well as FFA transport and utilization, may contribute to these rapid changes in plasma FFA levels. Adipose tissue lipolysis is higher in patients with OSA [38] and can be induced by exposure to hypoxia [12,19]. Although the timeline of changes in adipose tissue lipolysis and muscle FFA oxidation remains unclear, this study, combined with previously published data [38], demonstrates that increased adipose tissue lipolysis persists for at least several hours after awakening. In comparison, muscle lipid oxidation is not affected during comparable time periods in those with severe OSA because the tissue has already recovered from the hypoxia-induced suppression of lipid oxidation. This explanation is plausible for studies showing a rapid clearance of plasma FFA with a half-life of 3–4 min [62]; as reported by other researchers [63] and observed in our study, the explanation may also be plausible for observations of unchanged fasting FFA levels across a spectrum of OSA severity [63]. Furthermore, indirect calorimetry measurements performed shortly after awakening suggest an association between reduced lipid utilization and OSA severity [64]; however, whether muscle lipid oxidation is affected during sleep in OSA patients remains to be determined. Such a determination would require isotope techniques and a consideration of their limitations [62]. Sorting out the relative contribution of lipolysis to lipid oxidation has practical and important implications for the design of pharmacological interventions in OSA. A significant proportion of OSA patients do not tolerate the first-choice treatment option (i.e., continuous positive airway pressure therapy); additionally, these patients may be contraindicated for surgical treatment due to their health status [65,66,67]. These patients would strongly benefit from targeted pharmacological treatment, as demonstrated by the improvements in glucose metabolism after lipolysis inhibition in a mouse model of OSA [19].

Several limitations of the present study should be considered when interpreting and extrapolating the results. First, the study used muscle biopsies and analyzed lipid utilization and gene/protein expression ex vivo, without the (patho)physiological milieu of complex neuroendocrine regulation. However, muscle biopsies respond to known metabolic stimuli by increasing respiration in an ex vivo environment, and this method has been extensively used [68,69,70]. Notably, only FFA oxidation (O_2_ consumption) was investigated in this study; other aspects of lipid metabolism, such as lipid transport, storage, or de novo synthesis, were not analyzed. It should also be noted that three major players in FFA plasma and mitochondrial transport were investigated in this study; however, other proteins and FFA mechanisms were described but not analyzed [71,72]. Second, all measurements and studies were performed after awakening, providing several hours for recovery from OSA-related night factors. In contrast, these two limitations enable the identification of structural or long-term functional maladaptations (not only in muscle tissue) that persist during daytime hours and may represent potential drug targets. Third, our results should be interpreted and generalized cautiously due to the limited sample size and the fact that muscle biopsies and other metabolic parameters were investigated during wakening and not during sleep. Finally, OSA patients exhibit various phenotypes despite identical OSA severity as assessed by AHI, suggesting that other influential variables (e.g., sympathetic activation or stress/endocrine responses) play a role, and definitions based on AHI may not reflect the full extent of OSA [73].

In conclusion, the present study demonstrated that severe OSA did not modify muscle lipid oxidation, including the expression of essential regulatory proteins, in nondiabetic and T2DM individuals. We suggest that increased adipose tissue lipolysis, rather than reduced muscle FFA oxidation, is responsible for the reported elevations in plasma FFA levels in OSA patients and the resulting negative impact on glucose homeostasis. Consequently, adipose tissue metabolism may represent a plausible drug target for treating OSA-related metabolic derangements.

## 4. Materials and Methods

### 4.1. Participants

The participants were recruited through referrals from physicians and local media advertisements, as part of the larger FAMOSA study, as described previously [38]. The participants were recruited into four groups: nondiabetic persons with no or mild OSA (control, *n* = 14), nondiabetic persons with severe OSA (OSA, *n* = 9), patients with T2DM with no or mild OSA (T2DM, *n* = 10), and patients with T2DM and severe OSA (T2DM + OSA, *n* = 11). The inclusion criteria were age 18–85 years and a body mass index (BMI) of 22–40 kg/m^2^. T2DM was diagnosed according to the criteria of the European Association for the Study of Diabetes [74]. Notably, all patients with acute illness, decompensated chronic disease, or cardiac or renal insufficiency; as well as those treated with beta-blockers, corticoids, insulin, sulfonylurea, GLP-1 receptor agonists, and gliflozins; and those who had a body weight change of >5 kg over the last three months, were excluded. All participants provided written informed consent before participating in the study. The study was registered at ClinicalTrials.gov (NCT02683616) and approved by the Ethics Committee of Kralovske Vinohrady University Hospital, Prague (EK-VP/17/0/2014).

### 4.2. Sleep Study

The sleep study was performed using a type III device recording the hemoglobin saturation, heart rate, electrocardiogram, nasal airflow, and chest and abdominal respiratory efforts (Nox T3, Nox Medical, Reykjavik, Iceland) in the home setting. The acquired data were evaluated by a board-certified sleep physician according to the American Academy of Sleep Medicine criteria (apnea was defined as ≥90% reduction in airflow for at least 10 s, and hypopnea was defined as ≥30% reduction in airflow for at least 10 s with ≥4% desaturation). The severity of OSA was stratified by the apnea–hypopnea index (AHI): <5, no OSA; AHI ≥ 5 and <15, mild OSA; AHI ≥ 15 and <30, moderate OSA; AHI ≥ 30, severe OSA.

### 4.3. Biochemical Analysis, Clinical Investigations, and Muscle Biopsy

FFA in the serum was determined using the NEFA-HR2 assay (Wako Chemical Inc., Richmond, VA, USA). Other biochemical analyses were performed by the Institutional Department of Laboratory Diagnostics, Kralovske Vinohrady University Hospital, Prague. Patients visited the clinical research center after overnight fasting for metabolic and anthropometric assessments and a muscle biopsy. The measurements included biochemical analysis, blood count, coagulation, urinalysis, multifrequency bioimpedance measurements for body composition (Body Impedance Analyzer NUTRIGUARD-M, Data Input GmbH, Frankfurt, Germany), and measurement of the body weight, height, and waist circumference. Afterward, a frequent-sampling intravenous glucose tolerance test (IVGTT) was performed as previously described [38]. Briefly, two intravenous catheters were inserted into the antecubital vein, and basal sampling at −15, −10, −5, and −1 min was performed, followed by intravenous administration of 0.3 g/kg glucose at 0 min and 0.03 U/kg insulin (Humulin R, Lilly France S.A.S, Fegersheim, France) at 20 min. Blood samples for plasma glucose and insulin determination were collected at 2, 3, 4, 5, 6, 8, 10, 12, 14, 16, 19, 22, 24, 25, 27, 30, 40, 50, 60, 70, 80, 90, 100, 120, 140, 160, and 180 min. The values were subjected to a minimal model analysis [75] for insulin sensitivity (S_I_) and insulin secretion indices, including AIRg, disposition index (DI), S_I_, glucose effectiveness (S_G_), β-cell function, and insulin resistance.

Muscle biopsies were performed two weeks after the clinical investigation visit (to avoid the possible influence of the IVGTT test on muscle metabolism). Muscle biopsies were performed in a fasted state, and samples were taken from the lateral part of the vastus lateralis muscle of the dominant leg, approximately 10 cm above the knee, under aseptic conditions using a 5-mm Bergstrom needle, as described previously [76]. Fresh biopsy samples were immediately transferred into relaxing BIOPS buffer (10 mM CaK2-EGTA, 7.23 mM K2-EGTA, 20 mM imidazole, 20 mM taurine, 50 mM K-MES [2-(N-morpholino)ethanesulfonic acid], 0.5 mM dithiothreitol, 6.56 mM MgCl_2_, 5.77 mM ATP, and 15 mM phosphocreatine adjusted to pH 7.1). Biopsies were stored in BIOPS buffer on ice until the lipid oxidation was determined.

### 4.4. Lipid Oxidation Determination Using High-Resolution Respirometry

High-resolution respirometry adapted and validated for tissue homogenates obtained from muscle biopsy samples [77,78,79] was performed using an Oxygraph 2 K respirometer (Oroboros Instruments, Innsbruck, Austria). The analysis principle was based on the polarographic measurement of the oxygen consumption rate (OCR) of the analyzed sample using a Clark’s electrode. Before measurements, connective tissue, fat, and blood vessels were removed from the biopsy under a magnifying glass and subsequently homogenized by 4–6 strokes in an Elvehjem–Potter Teflon/glass homogenizer in 1 mL/100 mg biopsy sample in K media (10 mM Tris HCl, 80 mM KCl, 3 mM MgCl_2_, 5 mM KH_2_PO_4_, 1 mM ethylenediaminetetraacetic acid, and 0.5 mg/mL bovine serum albumin) at pH 7.4. Respirometry was performed at 30 °C without preoxygenation using 0.2 mL of 10% biopsy homogenate and 1.9 mL of K media. Analysis of the mitochondrial functional indices in homogenates was assessed after the addition of selected substrates to the respirometer chamber (using the Hamilton syringe) at 4-min intervals, according to the manufacturer’s recommendations, as follows (final concentration in the respirometer chamber): malate (1 mM) + ADP (1 mM) to evaluate baseline OCR using intracellular substrates, followed by administration of palmitoyl carnitine (0.04 mM) to assess lipid oxidation capacity, and administration of succinate (10 mM) to provide unlimited substrate for mitochondrial complex II to simulate maximal mitochondrial respiration. Data were normalized to the sample protein concentration (mg) determined using the bicinchoninic acid assay. All the chemicals were purchased from Sigma-Aldrich (St. Louis, MO, USA).

### 4.5. Protein and Gene Expression Quantification

qPCR: Biopsy samples were homogenized, and total RNA was extracted (TriPure Isolation Reagent, Roche Diagnostics, Rotkreuz, Switzerland) and treated with DNAse using a High Pure RNA Isolation Kit (Roche Diagnostics, Switzerland). A high-capacity cDNA Reverse Transcription Kit (Roche Diagnostics, Switzerland) was used to transcribe cDNA, which was subsequently assayed using a Real-Time PCR cycler ABI 750 (ThermoFisher Scientific, Waltham, MA, USA). TaqMan™ Fast Advanced Master Mix and probes Hs00192700_m1, Hs00169627_m1, and Hs03046298_s1 (Applied Biosystems, Carlsbad, CA, USA) were used to determine the expression of *FATP4* (fatty acid transport protein 4), *CD36* (platelet glycoprotein 4), and *CPT1* (carnitine-palmitoyl transferase I), respectively. Data were expressed relative to the geometric mean of *GUSB* (β-glucuronidase) and *TBP* (TATA box binding protein) gene expression (TaqMan probes Hs00427620_m1 and Hs00939627_m1, respectively) using the 2^−ΔΔCt^ method.

Western blot: Biopsy samples were homogenized in T-PER lysis buffer (ThermoFisher Scientific, USA), centrifuged (10.000 rpm, 10 min, 4 °C), and mixed with Laemmli buffer (1:1, Bio-Rad Laboratories, Hercules, CA, USA). SDS-PAGE was performed using 8% gels and blotted onto a 0.2 μm PVDF membrane for 2 h at 100 V in precooled transfer buffer (Bio-Rad Laboratories, USA). All membranes were blocked with 5% nonfat milk in TBS-T buffer (100 mM Tris-HCl, 150 mM NaCl, pH 7.6, 0.1% Tween-20) for 60 min. After washing with TBS-T, the membranes were incubated with the primary antibody overnight on a shaker placed in a refrigerator and subsequently washed in TBS-T. The following Abcam (Cambridge, UK) antibodies were used: anti-FATP4 (1:1000, Cat. No.: ab200353), anti-CD-36 (1:1000, Cat. No.: ab133625), anti-CPT1A (1:500, Cat. No.: ab234111), and anti-β-tubulin (1:3000, ab6046). A secondary antibody conjugated with HRP (1:10000, sc-2004, Santa Cruz Biotechnology, Dallas, TX, USA) was applied for 60 min; membranes were washed with TBS-T and subjected to chemiluminescence detection using a Radiance PLUS Chemiluminescent Substrate (Azure Biosystems, Dublin, CA, USA) and a ChemiDoc Imaging System (Bio-Rad, USA). Image Lab software (Bio-Rad, USA) was used for densitometric analysis, and band intensities of the evaluated proteins were normalized to β-tubulin signals.

### 4.6. Statistical Analysis

The differences in outcome variables between the groups (control, OSA, T2DM, and T2DM + OSA) were analyzed using one-way analysis of variance (ANOVA) with least significant difference post hoc tests; mixed-model analysis was employed for linear trend analysis. Correlations between the continuous variables were analyzed using Spearman’s correlation coefficients. GraphPad Prism 7 (GraphPad Software Inc., La Jolla, CA, USA) was used for the statistical tests and figure production. Statistical significance was set at *p* < 0.05 in all tests. Data are presented as means ± SEM.

## 5. Conclusions

The present study demonstrated that severe OSA did not affect muscle lipid oxidation, including the expression of essential regulatory proteins, in nondiabetic and T2DM individuals. We suggest that increased adipose tissue lipolysis, rather than reduced muscle FFA oxidation, is responsible for the reported elevations of plasma FFA levels in OSA patients and the consequent negative impact on glucose homeostasis. Consequently, adipose tissue metabolism may represent a plausible drug target for treating OSA-related metabolic derangements.

## Figures and Tables

**Figure 1 ijms-24-05308-f001:**
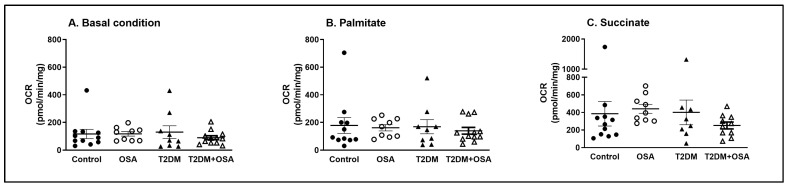
Oxygen consumption rate and substrate utilization in skeletal muscle biopsies. The oxygen consumption rate (OCR) in the muscle biopsies was measured using a respirometry chamber. Initially, the biopsy homogenate was supplemented with 1 mM malate + 1 mM ADP to evaluate the baseline OCR, defined mainly by the utilization of intracellular substrates (**A**). Subsequently, 0.04 mM palmitoyl carnitine was administered to assess lipid oxidation capacity (**B**), and 10 mM succinate was added to provide an unlimited substrate for mitochondrial complex II, thus maximizing mitochondrial respiration (**C**). Control (nondiabetic persons without sleep apnea syndrome), OSA (nondiabetic persons with severe sleep apnea syndrome), T2DM (type 2 diabetes mellitus patients without severe sleep apnea syndrome), and T2DM + OSA (type 2 diabetes mellitus patients with severe sleep apnea syndrome). Data are presented as individual values and error bars displaying the mean ± SEM; an ANOVA test was performed to assess differences between the groups.

**Figure 2 ijms-24-05308-f002:**
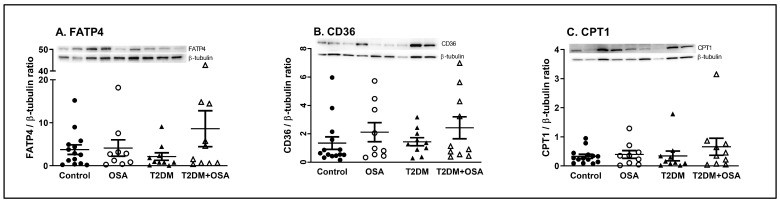
Protein expression of FATP4, CD36, and CPT1 in muscle biopsies. Western blot and densitometric analyses were performed to quantify the expression of (**A**) fatty acid transport protein 4 (FATP4), (**B**) platelet glycoprotein 4 (CD36), and (**C**) carnitine-palmitoyl transferase I (CPT1), relative to the expression of β-tubulin protein. Control (nondiabetic persons without sleep apnea syndrome), OSA (nondiabetic persons with severe sleep apnea syndrome), T2DM (type 2 diabetes mellitus patients without severe sleep apnea syndrome), and T2DM + OSA (type 2 diabetes mellitus patients with severe sleep apnea syndrome). Data are presented as individual values and error bars displaying the mean ± SEM; an ANOVA test was performed to assess differences between the groups.

**Figure 3 ijms-24-05308-f003:**
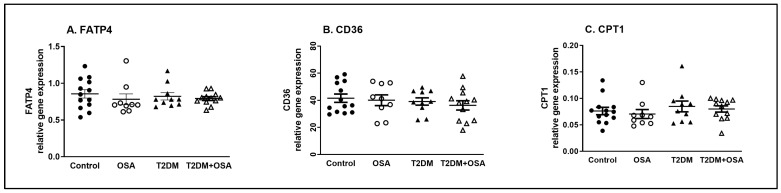
Relative gene expression of *FATP4, CD36*, and *CPT1* in muscle biopsies. Quantitative real-time PCR was employed to assess the gene expression of (**A**) *fatty acid transport protein 4* (*FATP4*), (**B**) *platelet glycoprotein 4* (*CD36*), and (**C**) *carnitine-palmitoyl transferase I* (*CPT1*), relative to the expression of the reference genes for *β-glucuronidase* and *TATA box binding protein* (geometric mean of the two). The relative gene expression values of all groups are expressed as 2^^ΔCT^. Data are presented as individual values and error bars displaying the mean ± SEM; an ANOVA test was performed to assess differences between the groups.

**Table 1 ijms-24-05308-t001:** Fundamental Anthropometric, Metabolic, and Sleep Characteristics of Subjects.

	Control (*n* = 14)	OSA (*n* = 9)	T2DM (*n* = 10)	T2DM + OSA (*n* = 11)
Men/Women	3/9	6/3	3/7	6/5
Age (years)	62.2 ± 1.3	63.4 ± 1.9	64.0 ± 2.1	63.4 ± 1.9
BMI (kg/m^2^)	33.5 ± 1.1	33.8 ± 1.0	32.7 ± 0.8	35.2 ± 1.0
Fat (%)	30.9 ± 2.6	28.1 ± 2.9	30.9 ± 2.6	30.6 ± 2.5
AHI	4.3 ± 0.7	47.2 ± 4.0 *	5.9 ± 0.9	47.8 ± 4.9 *^#^
ODI	4.3 ± 0.5	39.0 ± 3.4 *	6.8 ± 1.2	46.3 ± 5.1 *^#^
T90 (%)	1.8 ± 0.7	24.2 ± 7.6 *	10.1 ± 4.1	34.6 ± 9.2 *^#^
T85 (%)	0.2 ± 0.1	4.6 ± 2.3 *	0.4 ± 0.3	10.1 ± 4.3 *^#^
HBA_1_C (mmol/mol)	36.2 ± 0.1	38.4 ± 1.2	52.7 ± 3.5 *^†^	51.7 ± 3.4 *^†^
Glucose (mmol/L)	99.6 ± 1.0	102.7 ± 1.4	138.3 ± 6.1 *^†^	134.9 ± 4.7 *^†^
Insulin (mU/L)	10.3 ± 0.6	10.1 ± 0.3	14.4 ± 0.8	17.6 ± 0.6 *
Disposition index	1077.1 ± 88.5	652.1 ± 112.9 *	355.2 ± 109.5	174.2 ± 33.4 *^†^
AIRg (mU/L/min)	708.6 ± 63.1	507.4 ± 88.2	371.6 ± 112.0	152.0 ± 29.3 *^†^
IR (mM/mU/L^2^)	2.4 ± 0.1	2.4 ± 0.3	4.4 ± 0.3^†^	5.5 ± 0.5 *^†^
HOMA-IR	2.6 ± 0.2	2.6 ± 0.3	4.9 ± 0.3 *^†^	5.9 ± 0.6 *^†^
FFA (µmol/L)	530 ± 40	530 ± 50	500 ± 30	560 ± 40

Data are presented as mean ± SEM. BMI (body mass index), AHI (apnea-hypopnea index), ODI (oxygen desaturation index), T90 (percentage of total sleep time with hemoglobin oxygen saturation <90%), T85 (percentage of total sleep time with hemoglobin oxygen saturation <85%), HBA_1_C (glycated hemoglobin), AIR_g_ (acute insulin response to glucose), IR (insulin resistance index calculated from IVGTT), HOMA-IR (homeostatic model assessment for insulin resistance), and FFA (free fatty acids). * *p* < 0·05 compared with the control group (ANOVA with post hoc analysis), ^†^ *p* < 0·05 compared with the OSA group (ANOVA with post hoc analysis), ^#^ *p* < 0·05 compared with the T2DM group (ANOVA with post hoc analysis).

## Data Availability

The data presented in this study are available on request from the corresponding author.

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
