# Peer review of "Muscle Lipid Oxidation Is Not Affected by Obstructive Sleep Apnea in Diabetes and Healthy Subjects"

_ijms, 2023, doi:10.3390/ijms24065308_

Round 1

Reviewer 1 Report

This is an interesting article showed that severe  obstructive sleep apnea (OSA) is not associated with reduced muscle lipid oxidation and that metabolic derangements in OSA are not mediated through impaired muscle lipid oxidation. The strength of this study is defined by its study in human being. However, the following concerns reduce the enthusiasm for the acceptance of this manuscript.

1. The sample size is small.

2.  OSA is an disorder during sleep, it is necessary to assess muscle lipid oxidation during sleep in OSA patients.

Author Response

Authors thank the reviewer for manuscript assessment and valuable feedback on important issues. Authors believe that addressing reviewer`s comments helped to further improve the overall quality of the manuscript. 

1.The sample size is small.

Authors agree that limited sample size is a weak part of the study. On the other hand, 44 subjects were recruited in total, which represents significant efforts and resources invested in the study. Authors would also like to point out that methods used in this study are highly invasive (muscle biopsy) and demanding on the analytical part as well. To address this limitation, an additional text was added to the manuscript as described below.

2. OSA is an disorder during sleep, it is necessary to assess muscle lipid oxidation during sleep in OSA patients.

Authors completely agree with this concern, indeed, studying muscle metabolism during sleep would be closer to pathophysiological mechanism linking OSA with metabolic derangements. In fact, authors were considering multiple other approaches when planning this study, including the use of muscle microdialysis technique, employing isotope tracer/tracee methodologies or arierio-venous differences, however authors concluded that the (possible) disruption of sleep associated with practical clinical investigation using these techniques would significantly distort any observations and make interpretation of the results very ambiguous. Furthermore, it has been showed previously, that at least some of the impairments induced during sleep do persist during the wake state (e.g. sympathetic nervous system activation, impaired glucose metabolism or increased adipose tissue lipolysis). Finally, epidemiological evidence of increased risk of metabolic disorders associated with OSA also suggests that consequences of OSA do extend beyond the night period.      

To address issues raised by the reviewers, authors added the following text to the limitations in the Discussion part: ”Third, the study results should be interpreted and generalized with caution due to limited sample size and due to the fact, that muscle biopsies and other metabolic parameters were investigated in the wake period and not during sleep.” 

Reviewer 2 Report

Present manuscript entitled “Muscle Lipid Oxidation is Not Affected by Obstructive Sleep 2 Apnea in Diabetes and Healthy Subjects” that is very important study in current scenario because now a days sleep apnea is increasing very fast and it is basic root for cardiovascular disorders. As authors showed the molecular mechanism of related proteins. Although authors should must have include lipid profiles which give more meaning findings.    

The study looks promising. I would only suggest, if authors  could have discussed more in context to how these signalling molecules play important role in disease, it could have given a bigger and clearer picture to the researcher. The paper is technically sound, thoughtful and generally supports their conclusions. The statistical analysis is appropriate, and the paper is well placed in the context of the existing literature and field.

Over all, the manuscript is well written and after addressing the comments the current manuscript has significance for the publication in this esteemed journal.   

Author Response

Authors thank the reviewer for manuscript assessment and valuable feedback on important issues. Authors believe that addressing reviewer`s comments helped to further improve the overall quality of the manuscript. 

Authors have added a new paragraph discussing in broader view the role of key transport molecules investigated in the study. For more detailed information on the role of individual molecules, readers were referred to recent reviews. This comment partially overlapped with the concerns issued by another reviewer so the response to both issues is presented in a combination.

The following text was added to the Discussion: “The trio of investigated molecules: CD36, FATP and CPT1 represent key players in fatty acid transport from extracellular to intracellular space (mostly regulated by CD36 and FATP) and subsequently in FFA transport to mitochondria for oxidation (secured by CPT1). Dominant role of CD36 in FFA transport can be demonstrated by reduced transmembrane FFA transport after chemical CD36 inhibition [44,45], unfortunately, much less is currently known about the role and significance of FATP in sarcolemmal FFA transport [46]. Furthermore, CD36 was suggested to work in concert with CPT1 in mediating mitochondrial FFA transport/oxidation. Importantly, obesity, insulin resistance and type 2 diabetes [47] was associated with increased FFA muscle plasma membrane transport accompanied by elevated CD36 levels  [47,48] but also combined with reduced mitochondrial FFA oxidation [49–51] ultimately leading to increased muscle lipid accumulation. Hence, CPT1 muscle overexpression improved high-fat diet induced insulin resistance in rat [52][53]. For thorough reviews on the key players in muscle FFA transport and its consequences, readers are referred to exhaustive reviews [53–55].”

Reviewer 3 Report

In this article by Lattova and collaborators the correlation between obstructive sleep apnea (OSA) and lipid oxidation in muscle fibers in non-diabetic and dibetic OSA patients was investigated. The aim was to evaluate the possible correlation between muscle lipid oxidation and glucose metabolism abnormalities observed in OSA patients.

The experiments were conducted on a relatively small group of patients, but the results appear convincing, well presented and well discussed.

Possible limitations of the study are also clearly presented in the discussion.

Even if the possible link between many other dysregulated factors in OSA and type 2 diabetes (T2D) was clearly presented in the introduction, I suggest including a good and comprehensive review recently published in the Journal of Clinical Investigation (Hypoxia-Inducible Factors and Obstructive sleep apnea J Clin Invest.2020 Oct 1;130(10):5042-5051.doi:10.1172/JCI137560) reporting the possible correlation between hypoxia-induced factors and the development of T2D in OSA.

Furthermore, in the discussion, the author should include and discuss a rencent paper by Koenig et al. (J Endocr Soc . 2021 May 6;5(8):bvab082. doi:10.1210/jendso/bvab082. eCollection 2021 Aug 1. The Effect of Obstructive Sleep Apnea and Continuous Positive Airway Pressure Therapy on Skeletal Muscle Lipid Content in Obese and Nonobese Men) showing a relevant increase in intramyocellular and extramyocellular lipid muscle content (again in the vastus lateralis) in non-obese OSA patient. However, in the present article they do not detect alterations in the gene and protein expression of some enzymes involved in lipid transport and metabolism. How could they explain this discrepancy? How and why did they choose to evaluate the expression level of only the three (FATP4 and CD36 and CPT1) proteins and their relative gene expression? I think this may be the main limitation of this research work and it is worth explaining it better

Author Response

Authors thank the reviewer for manuscript assessment and valuable feedback on important issues. Authors believe that addressing reviewer`s comments helped to further improve the overall quality of the manuscript. 

  • Authors agree that limited sample size is a weak part of the study. On the other hand, 44 subjects were recruited in total, which represents significant efforts and resources invested in the study. Authors would also like to point out that methods used in this study are highly invasive (muscle biopsy) and demanding on the analytical part as well. To address this limitation, an additional text was added to the manuscript as described below.

To address issues raised by the reviewers, authors added the following text to the limitations in the Discussion part: ”Third, the study results should be interpreted and generalized with caution due to limited sample size and due to the fact, that muscle biopsies and other metabolic parameters were investigated in the wake period and not during sleep. ” 

  • Authors have added a reference to the comprehensive review to provide readers with a broader perspective on the topic. The reference was added to the Introduction section with the following text: Additionally, an important role of hypoxia-inducible factors (HIF) was also identified and summarized [25]“
  • Authors thank the reviewer for pointing out this interesting study, which was included in the Discussion together with a text as follows: Presented results are complementary to a recent study showing (in agreement with the present study) that OSA worsened glucose homeostasis, however, other authors also observed increased intra- and extra-myocellular lipid content in skeletal muscle. Furthermore, increased triglyceride accumulation and modified lipidomic profile in myocytes after hypoxic exposure was documented in vitro, presumably at least partly due to de-novo lipogenesis. These studies in concert suggest a hypothetical picture of unchanged lipid intracellular oxidation but increased lipid synthesis as a consequence of hypoxic exposure. Further studies are needed to address and quantify specifically and in a sufficient detail plasma membrane FFA transport under hypoxic conditions as extrapolation from gene/protein data on expression of key protein transporters is rather imprecise.”
  • Authors tried to better explain why the three transport molecules were assessed in this study – the reason is that these molecules represent key FFA transporters with extensive literature available and their role in pathogenesis of metabolic disorders at least partially established. As this concern is partially overlapping with an issue raised by another reviewer, the following text was added to the Discussion to address both concerns: “The trio of investigated molecules: CD36, FATP and CPT1 represent key players in fatty acid transport from extracellular to intracellular space (mostly regulated by CD36 and FATP) and subsequently in FFA transport to mitochondria for oxidation (secured by CPT1). Dominant role of CD36 in FFA transport can be demonstrated by reduced transmembrane FFA transport after chemical CD36 inhibition [44,45], unfortunately, much less is currently known about the role and significance of FATP in sarcolemmal FFA transport [46]. Furthermore, CD36 was suggested to work in concert with CPT1 in mediating mitochondrial FFA transport/oxidation. Importantly, obesity, insulin resistance and type 2 diabetes [47] was associated with increased FFA muscle plasma membrane transport accompanied by elevated CD36 levels [47,48] but also combined with reduced mitochondrial FFA oxidation [49–51] ultimately leading to increased muscle lipid accumulation. Hence, CPT1 muscle overexpression improved high-fat diet induced insulin resistance in rat [52][53]. For thorough reviews on the key players in muscle FFA transport and its consequences, readers are referred to exhaustive reviews [53–55].”
  • To provide more complete picture to the readers, authors also extended the limitations section of the Discussion and mentioned that the investigated molecules represent important, but not exhaustive, list of lipid transport molecules and mechanisms involved in muscle FFA transport/oxidation. The following text was added to the limitations section: “It should also be acknowledged that three major players of FFA plasma and mitochondri-al transport were investigated in this study, however, other proteins and FFA mechanisms were described but not analyzed in this study [61,62]”

Round 2

Reviewer 1 Report

The present study demonstrated that muscle lipid oxidation was not modified by the presence of severe OSA in nondiabetic and T2DM persons.  It is meaningful to investigate the molecular mechanisms linking obstructive sleep apnea (OSA) with type 2 diabetes. However, the limited sample sizes make the results not so convincing. It is good that the authors have tried to address these issues by adding an additional text

Author Response

Authors also submitted the manuscript for the second round of language revisions and reflected this in the manuscript Acknowledgement section by adding the text: Authors thank Thomas Secrest (Secrest Editing s.r.o.) and Elsevier Language Editing services for English language revisions.“

Reviewer 2 Report

Thank you so much for your revised version. Now manuscript is in publishable form.

Author Response

(The authors gave the same response as above.)
